# Reduction of Artifacts in Capacitive Electrocardiogram Signals of Driving Subjects

**DOI:** 10.3390/e24010013

**Published:** 2021-12-22

**Authors:** Tamara Škorić

**Affiliations:** Faculty of Technical Sciences, University of Novi Sad, 21000 Novi Sad, Serbia; ceranic@uns.ac.rs or tamara.ceranic@gmail.com; Tel.: +381-65-667-3311

**Keywords:** cECG filter, movement artefacts, binarized approximate entropy, KNN, DDNN

## Abstract

The development of smart cars with e-health services allows monitoring of the health condition of the driver. Driver comfort is preserved by the use of capacitive electrodes, but the recorded signal is characterized by large artifacts. This paper proposes a method for reducing artifacts from the ECG signal recorded by capacitive electrodes (cECG) in moving subjects. Two dominant artifact types are coarse and slow-changing artifacts. Slow-changing artifacts removal by classical filtering is not feasible as the spectral bands of artifacts and cECG overlap, mostly in the band from 0.5 to 15 Hz. We developed a method for artifact removal, based on estimating the fluctuation around linear trend, for both artifact types, including a condition for determining the presence of coarse artifacts. The method was validated on cECG recorded while driving, with the artifacts predominantly due to the movements, as well as on cECG recorded while lying, where the movements were performed according to a predefined protocol. The proposed method eliminates 96% to 100% of the coarse artifacts, while the slow-changing artifacts are completely reduced for the recorded cECG signals larger than 0.3 V. The obtained results are in accordance with the opinion of medical experts. The method is intended for reliable extraction of cardiovascular parameters to monitor driver fatigue status.

## 1. Introduction

The automotive industry has been making efforts to develop smart health systems, as part of supporting smart cars that can communicate with each other, transmit data to the cloud, and use smart e-health service systems [1,2,3,4,5]. Measurement of electrocardiograms (ECG), electroencephalograms (EEG), and respiratory activities, as well as assessment of the parameters of these time series in real-time, would contribute to the insight into the health condition of the driver while driving. The driver’s ECG, as well as the parameters derived from this signal, enable the assessment of alarming traffic situations such as driver fatigue [6], drowsiness [7], prediction of infarction development (of particular importance for the older group of drivers) [8], and EEG signal measurement contributes detection of driver fatigue [9], as well as prediction of emergency braking situations to activate the brake pedal when drivers are not able to react at the appropriate speed [10,11]. An additional motivation is the current demographic situation and the significant presence of older drivers in traffic. Ford has developed a car seat for heart rate monitoring (HR) [12]. Capacitive electrodes are installed in the car seat for non-contact (without direct contact with the skin) measurement of ECG signals through the driver’s clothing [1]. The main goal of this system was to monitor HR as an important parameter for assessing the health status of drivers [13], or the presence of drowsiness in drivers [14]. In addition to the capacitive electrodes that are built into the car seat, in Reference [15], a steering wheel covered with a conductive fabric-based dry electrode material is proposed, while Reference [5] describes a solution that additionally uses sensors built into the steering wheel and belt. The signals are sent via Bluetooth^®^ and processed on the built-in computer. A car seat equipped with Internet of Things (IoT) sensors for measuring ECG and EEG signals with a suitable transmitter that sends physiological data to the database for further processing and prediction of the driver’s health has been developed [3,4].

Limiting factors are specific conditions in which signal measurements are performed, i.e., recording while driving in a car, measurement without direct contact with the driver (recording over clothes made of different materials), and the dominant presence of movements, as well as the distractions while driving. The measured signals have very small amplitudes, with the dominant presence of artifacts that represent an obstacle to a reliable assessment of health parameters.

The presence of artifacts in the signals, as well as the estimation of parameters without a priori signal processing, can lead to the generation of false alarms or isolation of false outliers, which is becoming a relevant topic in database processing [16]. Recording by a capacitive electrode in clinical conditions, while the subjects were sitting still, was analyzed to determine the possibility of clinical application [17]. Detection of cardiac arrhythmia, however, is not reliable due to the impossibility of reliable identification of P and T waves, mostly due to the presence of movement artifacts [17]. Therefore, medical experts have pointed out that it is necessary to improve the algorithm for the reduction of artifacts in ECG signals recorded by capacitive electrodes [17]. For this reason, various efforts are being made to reduce artifacts. The initial proposal was to eliminate all beats that would have atypical values (below 30 beats/min or over 120 beats/min), i.e., these signal segments would be treated as artifacts [18]. The authors themselves stated that this very simple approach is not good enough for a realistic estimate of the signal parameters [18]. In addition, this approach removes the parameters that indicate the occurrence of unwanted pathologies. In Reference [1], an algorithm for artifact reduction based on principal component analysis (PCA) was proposed, with mandatory pre-processing of data to construct an appropriate training data set for PCA. The use of a training data set with a dominant presence of artifacts could lead to inaccurate classification by PCA, so pre-processing of signals recorded while driving is a mandatory step [1]. Rough signal preprocessing was performed based on the QRS complex quality assessment to reduce the number of artifacts in the data. QRS complexes were detected using open-source ECG analysis (OSEA) [19], and values that were many times higher than expected (so-called “outliers”) were categorized based on a threshold representing the product of parameters describing the amplitude and standard deviation of QRS complexes [1]. After pre-processing of the data, the final elimination of the artifacts was performed based on the calculated eigenvalues, and, for that purpose, the threshold based on the Hotelling T square value, described in Reference [20], was used. The authors pointed out that the algorithm needs to be improved because it also eliminates QRS complexes of small amplitudes compared to noise [1].

One of the available algorithms for the assessment and detection of cECG signal artifacts is based on the injected signal and modeling of the capacitive system [21]. The model has been tested only in laboratory conditions and in simulations [21], so further confirmation in real systems is necessary [22]. Hardware upgrades have also been proposed [22], i.e., it is necessary to install an additional electrode that would provide motion information to form a reference signal for active noise cancellation (ANC), including an adaptive filter. There are many methods developed for the reduction of artifacts for the traditional measurement of ECG signals with direct contact with the patient’s body and with the application of a gel. In the review paper of Reference [23], these techniques are categorized as techniques based on Empirical mode decomposition (EMD), wavelet transforms, hybrid models, deep-learning models, and various filters. However, for artifacts on cECG signals, EMD and wavelet-based techniques did not yield adequate results due to the dominant presence of large amplitudes and irregular motion noise characteristics, which was also observed in Reference [22]. Hopes have been pinned on a modified noise reduction method, Detrended Fluctuation Analysis (DFA)-EMD, based on the idea that, in addition to decomposing signals into intrinsic mode functions (IMF), the DFA parameters for each IMF should be estimated to characterize the IMF, as a carrier noise or information [24]. In Reference [25], however, it was stated that the DFA-EMD method failed to mitigate the presence of artifacts in the cECG signal recorded while driving, due to the dominant presence of very large amplitudes of the artifacts.

The aim of this paper is to develop a method for reducing artifacts from ECG signals recorded by capacitive electrodes (cECG) in moving subjects. The basic idea of the proposed method is that quantified signal fluctuation around a linear trend of fixed-length segments can detect segments with artifacts. It is assumed that segments with a very high or very low value of fluctuations correspond to segments with coarse and slowly changing artifacts (frequency range from 0.5 to 15 Hz), respectively. We expect that the assessment of an acceptable level of fluctuation will indicate coarse and slowly changing artifacts and enable their elimination. Then, the reliability of cardiovascular parameters estimated from cECG signals without artifacts would increase, as well as the accuracy of classification techniques.

## 2. Materials and Methods

### 2.1. Materials

Acquisition of cECG time series while driving was performed using 6 electrodes built into the car seat. The electrodes are arranged on three levels of two electrodes in the upper part of the car seat (a detailed photograph is given in References [1,18]). Three electrodes with the most reliable measurements were manually selected [18]. The process of recording the cECG signal involved 6 volunteers (male, aged 39.8 ± 26.2 years) who drove in the city (about 2 h of recording), on the highway (about 8.8 h of recording), and the polygon in Belgium (about 2.5 h of recording) [18]. Thirty-one measurements were performed, comprising three *cECG* signals (designated as *cECG_1_* = *Electrode_1_*
−
*Electrode_2_*, *cECG_2_* = *Electrode_2_*
−
*Electrode_3_*, and *cECG_3_* = *Electrode_3_*
−
*Electrode_1_*), as well as the reference ECG signal. Thus, each measurement while driving, whether driving on a highway, polygon, or city, provided four simultaneously recorded signals: *cECG_1_*, *cECG_2_*, *cECG_3_*, and a reference signal, a total of 124 signals. The reference signal was measured using the equipment of the biosignal amplifier g. Bsamp from g.tec medical engineering GmbH Schiedlberg, Austria (details in Reference [26]) [18]. An A/D converter (NI-USB6259) was used for capacitive electrodes (cECG_1_, *cECG_2_*, and *cECG_3_*) and reference measurements, with an amplitude resolution of 16 bits per sample and sampling frequency of 1000 Hz [1], except for two signals with a sampling frequency of 200 Hz [18].

The *cECG* time series, recorded while lying on the bed, was performed using 12 built-in electrodes in the bed, from which three ones with the highest quality of measurements were automatically selected [18]. *cECG_1_*, *cECG_2_*, and *cECG_3_* were formed in the same way as measurements recorded during driving by three automatically selected measurements. Ten volunteers (aged 27.8 ± 4.3 years) participated in the recording procedure. They moved according to a certain protocol to simulate movements during sleep and generate coarse artifacts [18]. In the first half of the measurement, the volunteers moved every 60 s, while, in the second half of the measurement, they were asked to lie for 120 s, then to move for 60 s, and then lie down again for 120 s [18]. In this experiment, the sampling frequency was 400 Hz. The reference signal was measured by an MP70 (details in Reference [27]) of Philips (Eindhoven, the Netherlands) [18].

The database with all experimental recordings is publicly available [28], and the experiment is described in detail in Reference [18]. All volunteers gave written consent [18].

Duration of recording and amplitudes value of cECG time series, publicly available [28], are shown in Table 1 and Table 2.

Table 1 shows the mean values of the absolute signal amplitudes ± standard deviation (SD) recorded capacitive electrodes. It is noticeable that the intensity of recorded cECG signals while driving is very low, while its duration is very large (Table 2). *cECG**_1_*, *cECG**_2_*, and *cECG**_3_* have the same length because they are recorded simultaneously, while driving or moving on the bed.

Manual notation of *R* peaks (maximum in the *QRS* complex) was performed by two medical experts independently. If there was a difference of opinion, the *R* peaks were marked as *NaN* [18]. Results of peaks detection by OSEA software (described in Reference [19]) were analyzed in Reference [18] and are available in Reference [28].

Examples of the raw cECG signals from two sets are shown in Figure 1a,b. Red markers point to the annotated R peaks according to medical experts. The original purpose of the labeled R peaks was to assess the accuracy of algorithms for automatic detection of R peaks in cECG, such as OSEA software, and the possibility of reliable estimation of HR [1,18]. The results confirmed the possibility of a reliable assessment of HR while driving in the case of eliminating time intervals with artifacts [1]. Our aim is to eliminate the artifacts and prepare the signal for further analysis. The beat-to-beat HR time series is one of the targets.

Two types of artifacts are distinguished: coarse artifacts, which occur as amplitude peaks but different amplitude values (examples are marked by green rectangles in Figure 1a,b), and slow-changing artifacts that are very similar to the useful part of a signal (examples are marked by blue rectangles in Figure 1a–c). Figure 1c,d show enlarged useful segments that medical experts have marked while driving and lying on the bed, respectively. The signal segment is treated as useful if R peaks can be detected by medical experts, or as useless if the R peaks, due to artifacts, cannot be detected. There are no manually marked artifact types in the publicly available database [28], neither coarse nor slow-changing artifacts.

After visual inspection, we noted that the cECG time series also differs in the number of coarse artifacts. Figure 2 shows the cECG time series (a) with a moderate amount of artifacts and (b) without coarse artifacts, while Figure 1a corresponds cECG time series with a very large amount of coarse artifacts. In accordance with the opinion of medical experts, useful parts of the cECG time series have been marked in red. Table 3 shows the number of recordings for each observed group during driving. The *cECG_3_* time series is not included in the analysis because there is no publicly available notation of the useful part of the signal. The total number of cECG time series per groups is small, but the long duration of recording (Table 2) enabled reliable signal analysis. Unfortunately, it is noted that a few of the cECG time series are into groups of cECG with a moderate amount of artifacts, and without artifacts, as a consequence of recordings conditions during driving. *cECG_1_*, *cECG_2_*, and *cECG_3_* time-series recorded while lying on the bed belong to a group of cECG with a large amount of artifacts due to moving subjects according to the protocol (total number of cECG is 60).

The signal-to-noise ratio is expressed as SNR = $10\cdot \log\left(\frac{P_{s}}{P_{n}}\right)$, where $P_{s}\;$corresponds to the power of cECG time series comprising only the useful segments according to the expert opinion, and $P_{n}$ corresponds to the power of all segments that were declared as useless. The signal segment is treated as useful if R peaks are labeled, or as useless if the R peaks are not labeled by medical experts. Its mean value is equal −40.01 ± 33.64 dB for *cECG**_1_*, and −30.99 ± 23.39 dB for *cECG_2_*, during driving. The power of time series is estimated as the sum of squared amplitude divided by length of time series. The negative value of SNR is a consequence of very high amplitude values of the coarse artifacts if compared to the useful parts of the signal. The high SD is a consequence of the different amounts of coarse artifacts in time series.

We also analyzed the power distribution over the frequency bands of slow-changing artifacts in *cECG**_1_* and *cECG**_2_* during driving. The power distribution averaged over ten signals is shown in Figure 3. The signals comprise segments with slow-changing artifacts manually extracted from *cECG**_1_* and *cECG**_2_*. The ECG spectral components (0.05~150Hz), on the other hand, are mainly concentrated in the range of 0.05~35 Hz [29] so that the spectral overlap is observed, especially, for the frequency less than 15 Hz. For this reason, classical band-pass filters cannot be implemented for artifact removal [30,31].

### 2.2. Reduction of Artifacts Based on Fluctuation in cECG Time Series

The estimation of signal segment fluctuation was performed following the first part of DFA procedure [32]. The samples of the *cECG* time series *x* of length *N* are denoted by $x_{k},\;k=1,\ldots N$. In the first step, a vector of cumulative sums *Y*(*i*)*, i* = 1,..., *N* is formed from the elements obtained by summing *i* successive centralized samples of the time series *x* [32]:(1)$$Y\left(i\right)=\sum_{k=1}^{i}\left[x_{k}-\langle x\rangle \right]\;i=1,\ldots \ldots ,N,$$
where the *k*-th centralized sample is formed by subtracting the mean value of the time series 〈x〉 from the *k*-th sample $x_{k}$.

The vector of cumulative sums Y is divided into non-overlapping segments of the same length [32]. Since the length of the vector of cumulative sums *N* need not be divisible by the number of segments, the last segment is usually shorter and needs to be omitted. This is not a problem, as the time series are very long (Table 2), so removing one segment would not compromise the reliability of the results. In the case of short time series, the segmentation procedure is repeated twice, the first from the beginning, and then starting from the end of the time series. The total number of observed segments is doubled. In this way, the reliability of result would not be compromised. The short time series is defined as series with less than 10,000 samples [33]. In the case of driving, it is only 10 s of recording. So, the procedure of repeated segmentation from the other end of time series is not applied.

The vector of cumulative sums *Y*(*i*), i=1,……,N, is divided into *N/SL* segments of length *SL*. The segments are denoted as $\Upsilon_{j}\left(k\right),\;j=1,\ldots ,\frac{N}{SL}$, where *k* denotes the samples within a particular segment, k=1,…,SL. Each segment is approximated by polynomial of the *v*-th order $p_{j,v}\;$that represents the trend of segment number *j*. Subtracting a trend from a segment leads to a detrended segment [32]:(2)$$\Upsilon_{j,SL}\left(k\right)=\Upsilon_{j}\left(k\right)-p_{j,v}\left(k\right),j=1,\ldots ,\frac{N}{SL},k=1,\ldots ,SL,$$
and
(3)$$p_{j,v}\left(k\right)=a_{v}\cdot \Upsilon_{j}{}^{v}\left(k\right)+a_{v-1}\cdot \Upsilon_{j}{}^{v-1}{\left(k\right)}_{}+\ldots +a_{0},\;$$
where $a_{v},a_{v-1},\;a_{0}—\mathrm{polynomial}\;\mathrm{coefficients}\;\mathrm{on}\;a\;\mathrm{segment};\nu —$polynomial order.

The most common polynomial that is used for this method is linear (*v* = 1) [34], so we implemented linear approximation.

The detrended fluctuation analysis function $F_{D}{}^{}\left(j\right)\;$of one segment of the time series is calculated as the sum of the square value of the difference between the original value of the time series and the trend of a given segment divided with SL [32]:(4)$$F_{D}{}^{}\left(j\right)\;=\sqrt{\frac{1}{SL}\cdot \sum_{k=1}^{SL}\left\{\;\Upsilon_{j,SL}^{2}\left(k\right)\right\}},j=1,\ldots ,\frac{N}{SL}.$$

The basic idea of the proposed method is to use the estimated value of fluctuation $F_{D\;}$ for the artifacts reduction. The $F_{D\;}$ values of extracted part of raw *cECG* signals recorded while driving is presented in the upper panel of Figure 4. Figure 4 shows an example of an extracted part of raw cECG from an available database, which is intended to show isolated characteristic segments for analysis. As expected, larger value of $F_{D}{}^{}\left(j\right)$ was obtained for the segments with coarse artifacts, due to the expected larger deviation from the linear trend of segments. However, we should be careful about establishing criteria for an acceptable level of fluctuation, as the detrended fluctuation $F_{D}{}^{}\left(j\right)$ of useless segments may be comparable to the correct segments. The range of $F_{D}{}^{}$ value depends on the time series, as described in detail in the next section.

Figure 4 shows that R peaks detected by OSEA software (marked by black rectangles) are in accordance with experts’ opinions in useful segments. In addition, the amplitudes of the useful parts are very small compared to the coarse artifacts.

Unfortunately, the amplitudes of the useful parts are comparable to the parts of the signal with slow-changing artifacts in which the cECG was not detected. One example of such a case is isolated in Figure 1c. Figure 1c shows enlarged part of useful segments of the signal shown in Figure 1a, marked by brown a rectangle, and the useless segment by blue rectangle of comparable amplitude. So, artifact reduction cannot be performed on amplitude values alone.

Appropriate cECG pretreatment, shown in Figure 1, would contribute to greater accuracy of R peak detection algorithms, as well as parameters derived from cECG. So, it presents the first step in developing an auxiliary tool that, based on the parameters extracted (from other sources, not only cardiovascular), would detect possible fatigue in the driver and trigger an alarm that would warn him.

#### Method for Artifacts Reduction

In the first step of the algorithm, the time series *x* should be divided into a non-overlapping segments of length SL. $F_{D}{}^{}\;$is estimated according to Equation (4) for each segment.

Before artifact reduction, it is necessary to check the presence of coarse artifacts in the time series. Namely, the existence of the coarse artifacts is not known a priori. The parameters that influenced the formation of the criteria for checking the presence of coarse artifacts are the maximum and minimum value of time series *x* (max(*x*), min(*x*), respectively), and the square root of the second moment *M*. Comparing time series with a large amount and the moderate amount of coarse artifacts to time series without coarse artifacts (examples are given in Figure 1a, Figure 2a,c), we note that the difference between the maximum and minimum value of time series *x*, max(*x*)-min(*x*), was larger for signals with coarse artifacts compared to signals without coarse artifacts. Subtracting the square root of the uncentralized second moment *M,*
$M=E\left(x^{2}\right)=\frac{1}{N}\sum_{k=1}^{N}x_{k}^{2}$*,* from this difference made it possible to distinguish between signals with a large amount of coarse artifacts and a moderate amount of coarse artifacts, as the value of *M* is larger for time series with more coarse artifacts.

The final value of criterion for the presence of coarse artifacts was determined experimentally:(5)$$(\left(\max\left(x\right)-\min\left(x\right)\right)/2)-\sqrt{M}>1,\;M=E\left(x^{2}\right)=\frac{1}{N}\sum_{k=1}^{N}x_{k}^{2},$$
with max(*x*), min(*x*)-maximum, and minimum value of time series *x*, respectively, and *M-* the second moment of time series *x*.

Figure 5 shows the value of Equation (5) for all three observed groups of cECG (Table 3). The value of Equation (5) is lower than 1 for signals without the presence of coarse artifacts (below the gray line in Figure 5).

For a large and moderate groups, the value of Equation (5) is larger than 1, with a notable distinction between these two groups. All the cECG time series recorded during lying on the bed fulfilled the condition for the presence of a large amount of artifacts (Equation (5)), as expected (Figure 5b).

If the condition of Equation (5) is fulfilled, artifacts from time series should be reduced. To reduce the presence of coarse and slow-changing artifacts, we have developed a set of formulae for automatically estimating the level of detrended fluctuation of time series segments as the criteria for the useful or useless segments.

The first threshold $TH_{1}$ is intended to reduce the coarse artifacts.

Threshold value $TH_{1}$ is equal to:(6)$$TH_{1}=\left(\sqrt{{\left(\left(\max\left(x\right)-\min\left(x\right)\right)/2\right)}^{2}-M}\right)\cdot C\cdot median\left(F_{D}\right)+\frac{\left(SD\left(F_{D}\right)+SD\left(x\right)\right)}{SD\left(F_{D}\right)\cdot SD\left(x\right)}\cdot C_{1},$$
where $median\left(F_{D}\right)$ is the median value of detrended fluctuation function of all segments in time series, $SD\left(F_{D}\right)$ and SD(x) are the standard deviation of $F_{D}$ and *x*, respectively, max(*x*), min(*x*)-maximum, and minimum value of time series *x*, respectively, and *M-* the second moment, while value *C* is constant value from the range, C∈{0.15–0.35} $\frac{1}{V^{2}}$ and $C_{1}=1\;V$.

Figure 6a shows $median\left(F_{D}\right)$ for cECG record during lying on the bed (gray) or driving in the car (dark green). The median value is higher for signals with a large presence of artifacts (marked by filled squares) compared to signals with moderate (marked by unfilled squares) or no artifacts (marked by unfilled triangle) in both signal groups. The obtained results are in accordance with the expectations motivated by Figure 4, where higher values of $F_{D}\left(j\right)$ were noticed in the segments in which coarse artifacts are present. In Figure 6b,c, we note that the values of $SD\left(F_{D}\right)$ and *SD*(*x*) for signals with a moderate amount of artifacts or without artifacts are smaller compared to the standard deviation of time series with a large amount of coarse artifacts, which is also in line with expectations. $SD\left(F_{D}\right)$ and *SD*(*x*) have a larger impact on the final value of $TH_{1}$ value, while the influence of the $median\left(F_{D}\right)$ is negligible for the signal with a moderate amount of artifacts. In the case of signals with a large amount of coarse artifacts, the influence of $median\left(F_{D}\right)$ is larger compared to $SD\left(F_{D}\right)$ and *SD*(*x*).

The threshold evaluation includes an empirical parameter *C*. To find the most suitable value, we analyzed the percentage of preserved *R* peaks and the percentage of eliminated useless parts of the signal that might generate false. The results are presented in Figure 7, for the range of values *C* ∈ {0.05 to 1}, and for the segment length *SL* = 0.5 s. The gray rectangle with *C* values from 0.15 to 0.35 indicates the range of values for which the best performances are achieved. Visual inspection of the cECG after the reduction of artifacts can determine the presence of coarse artifacts. We used strict criteria to assess the presence of coarse artifacts, and a time series with at least one coarse artifact is treated as a time series in which coarse artifacts are not successfully reduced. Within this range, 90% to 97% of useful signal parts are preserved, and 96% to 100% coarse artifacts are eliminated. The recommended value is the median point, *C* = 0.25. *R* peak annotations were available for *cECG* groups recorded while driving and in bed.

After elimination of all segments (i.e., excluding segments from further analysis) that fulfilled condition $F_{D}\left(j\right)>TH_{1}$, we check adjacent segments for the possibility of coarse artefacts partially spilling over the adjacent segments (an example of such segments is marked with asterisks in Figure 4). To be on the safe side, $F_{D}\left(j\right)\;$values of adjacent segments are compared with the $TH_{2}$.
(7)$$TH_{2}=\frac{TH_{1}\;}{2}.$$

If $F_{D}\left(j\right)\;$values exceed half of $TH_{1}$ value, this segment is declared as useless, and it is eliminated from cECG.

The third threshold, $TH_{3}$, has a role to eliminate segments with very small deviation from linear trend, i.e., slow-changing artifacts. If the square difference between the value of the sample, and the estimated trend of sample is equal to or less than 0.01, and if this condition is fulfilled for all samples in the segment, that segment is not treated as a carrier of useful information. In that case, (Equation (4)) is equal to $F_{D}{}^{}\left(j\right)=\sqrt{\frac{1}{SL}\cdot SL\cdot 0.01}$ = 0.1, so
(8)$$TH_{3}=0.1.$$

The problem of elimination of this type of artifact is expressed in cECG recorded in cars, where the signal intensity is very low (Table 1), so, in this way, it is possible to eliminate significant parts of the useful signal (compare part of cECG marked by blue and red rectangles in Figure 1c). For these reasons, additional protection was introduced, and the comparison with the $TH_{3}$ threshold is made only if the difference between the mean value of $F_{D}$ and$\;SD\left(F_{D}\right)$ is greater than $TH_{3}$. In this way, the possibility of an incorrect elimination of the useful signal segments is reduced. In the database [28], there is no manual notation of slow-changing artifacts, and, since they are comparable to a useful part, it is difficult to be identifiable visually. For these reasons, the success in eliminating slow-changing artifacts is observed by the overall accuracy of the algorithm, i.e., by comparing the useless segments according to the assessment of the algorithm with the useless segments marked by the opinion of medical experts.

The pseudocode explaining Algorithm 1 is shown below.
**Algorithm 1:** Reduction of Artifacts in *cECG* Time SeriesInput: *cECG* time series1. Divide the signal into non-overlapping segments *SL* = 0.5 s.2. Estimate the fluctuation for each segment3. Form array *F_D_* from estimated fluctuation for each segment4. Determine: the minimum value *cECG_MIN_*, maximum value *cECG_MAX_*, and the second uncentered moment *M*5. Calculate *TH*_1_6. Calculate *TH*_2_7. Calculate *TH*_3_8. **if** the difference between difference of half *cECG_MAX_* and *cECG_MIN_* and square root of *M* is less than 1            **if** element in array *F_D_* is larger than *TH*_1_ do            eliminate observed segment in *cECG*            **if** difference between current element in array *F_D_* and next element larger than *TH*_2_                 eliminate next segment in *cECG*            end if            **if** difference between current element in array *F_D_* and previous element larger than *TH*_2_                 eliminate previous segment in *cECG*            end if             end if            end if 9. Calculate the mean value of *F_D_* and standard deviation of *F_D_*10. **if** difference between the mean value of *F_D_* and *SD*(*F_D_*) is larger than *TH*_3_
**do**            **if** element in array *F_D_* is less than *TH*_3_
**do**            eliminate observed segment in *cECG*            end if            end if             Output**:**
*cECG* time series with reduced artifacts

### 2.3. Binarized Entropy (BinEn)

We also analyzed a method that does not require artifacts removal. Such methods are rare, almost non-existent. Binarized entropy (BinEn) [35] is one of them, developed for another harsh environment—mobile crowdsensing systems—where the reduction of artifacts is not feasible. A brief recapitulation of BinEn adapted to a single data set is below.

In the first step, time series x are binary differentially encoded and split into *m*-sized binary vectors [35]:$$c=\left\{\begin{array}{c} 0\;x_{i+1}-x_{i}\leq 0\\ 1\;x_{i+1}-x_{i}>0 \end{array}\;\right\}\;i=1,\ldots \ldots \ldots N$$
(9)$$C_{m}^{i}=\left[c_{i},c_{i+\tau },\ldots ,c_{i+\left(m-1\right)\cdot \tau }\;\right],\;i=1,\ldots ,N-\left(m-1\right)\cdot \tau ,\;$$
where the delay *τ* is distances the elements of the vector from each other, and *m* is size of vectors. In most applications *τ* = 1, m∈{1, 2, 3, 4} [35]. In the BinEn, the vectors are binary, so the number of different vectors is $2^{m}$, and each vector can be assigned a decimal number k [35]:(10)$$k=\overset{m-1}{\underset{n=0}{\sum }}c_{i+n\cdot \tau }\cdot 2^{n}.$$

$N_{C}^{\left(m\right)}\;$represents the number of occurrences of a certain vector series in the observed time series C [35]:(11)$$\;N_{C}^{\left(m\right)}\left(k\right)=\sum_{i=1}^{N-\left(m-1\right)\cdot \tau }I\left\{\sum_{l=0}^{m-1}c_{i+l\cdot \tau }\cdot 2^{l}=k\right\},\;k=0,\;1,\;\ldots ,\;2^{m}-1.$$

I{}—indicator function equal to one if the condition is met, and zero otherwise.

The estimation of probability mass function of observed vectors in C is equal to:(12)$${\hat{P}}_{C}^{\left(m\right)}=\frac{N_{C}^{\left(m\right)}\left(k\right)}{N-\left(m-1\right)\cdot \tau }\;.$$

In the following step, it is necessary to find the distance d between each pair of vectors. Distance d is calculated according to the Hamming distance [35]:$$d\left(C_{m}^{i},\;C_{m}^{j}\right)=\sum_{k=0}^{m-1}c_{i+k\cdot \tau }⊕c_{j+k\cdot \tau }=\sum_{k=0}^{m-1}I\left\{c_{i+k\cdot \tau }\neq c_{j+k\cdot \tau }\right\}$$
(13)i,j=1,…,N−(m−1)·τ, 
where ⊕ notes ex-or logic function, and *I*
{.} indicators function. The distance $d\left(C_{m}^{i},\;C_{m}^{j}\right)\;$between the vectors is a discrete variable that can have one of the m+1 values, that is, $d\left(C_{m}^{i},\;C_{m}^{j}\right)\in \left\{0,1,\ldots m\right\}$ [35].

The matrix of Hamming distance is denoted by *H*(*m*) [35]. Elements of matrix H are the distance between the vector whose decimal represents *k* and the vector whose decimal represents *n,* ($h_{k\cdot n}^{}$) [35]. The probability that vector $C_{m}^{i}$ occurs in C is estimated based on the value in matrix H, which gave information on which vectors are in distance less than *r* from $C_{m}^{i}$, and Equation (11) that gave information about a number of vectors that are at the same distance [35]:(14)$$\begin{array}{cc} {\hat{p}}_{k}^{m}\left(r\right) & =Pr\left\{d\left(C_{m}^{i},\;C_{m}^{}\right)\leq r\right\}\\ & =\frac{1}{N-\left(m-1\right)\cdot \tau }\cdot \overset{2^{m}-1}{\underset{n=0}{\sum }}N_{C}^{\left(m\right)}\left(n\right)\cdot I\left\{h_{k\cdot n}^{\left(m\right)}\leq r\right\}\\ & =\sum_{n=0}^{2^{m}-1}{\hat{P}}_{C}^{\left(m\right)}\left(n\right)\cdot I\left\{h_{k\cdot n}^{\left(m\right)}\leq r\right\}. \end{array}$$

In the next step, value of summand $\hat{\Phi }$ is calculated as average of logarithm ${\hat{p}}_{k}^{m}$ [35]:(15)$${\hat{\Phi }}^{m}\left(r,N,\tau \right)=\frac{1}{N-\left(m-1\right)\cdot \tau }\cdot \overset{2^{m}-1}{\underset{k=0}{\sum }}N_{C}^{\left(m\right)}\left(k\right)\cdot \ln\left({\hat{p}}_{k}^{m}\left(r\right)\right)=\sum_{k=0}^{2^{m}-1}{\hat{P}}_{C}^{\left(m\right)}\left(k\right)\cdot \ln\left({\hat{p}}_{k}^{m}\left(r\right)\right).$$

The final value of BinApEn is estimated on model of approximate entropy (detail in Reference [36]) [35]:(16)$$BinApEn\left(m,r,N,\tau \right)={\hat{\Phi }}^{m}\left(r,N,\tau \right)-{\hat{\Phi }}^{m+1}\left(r,N,\tau \right).$$

*BinSampEn* is a binarized version of sample entropy (proposed in Reference [37]), which excludes self-similarity (comparison of vectors with themselves) [35]:(17)$$BinSampEn\left(m,r,N,\tau \right)=-10\cdot \log\left(\frac{{\hat{\Phi }}^{m+1}\left(r,N,\tau \right)}{{\hat{\Phi }}^{m}\left(r,N,\tau \right)}\right).\;$$

### 2.4. Classifiers

The K nearest neighbors (KNN) algorithm is a simple supervised machine learning algorithm that classifies data based on estimates of *K* the nearest neighbors [38]. The K nearest neighbors are found by the distance between test and training objects in feature space [38]. The test object is classified into the appropriate class, in which the majority of K neighbors belong [38]. We used Euclidean distances to determine K nearest neighbors, the number of observed classes is two (driving in the city and driving on open roads). The number of K is selected by cross-validation of 10% of training data sets (K = 5).

The Deep Dense Neural Network (DDNN) is a deep learning technique that includes an input, output layer and a fully connected layer between those two layers [39]. We used model custom architecture (15 fully connected layers with 128 units, followed by drop layer output). The proposed architecture is very simple due to the existence of only two classes and a small database (details described in Section 2.5). We used an Adam optimizer [40] during DDNN training and cross-entropy as loss function [41]. The drop layer output is used for the regularization procedure to reduce overfitting to the training data set. The purpose of the experiment was to test the sensitivity of the classifier to the presence of artifacts in the signal.

### 2.5. Statistical Analysis

Some of the illustrative results are presented as graphs showing mean ± standard deviation. Statistical significance between observed groups was checked by *t*-test for paired samples in MATLAB R2013a. We used significance level *p* < 0.01 for all compared groups.

To form a database of appropriate sizes for testing and training KNN and DDNN, we divided the *cECG_2_* time-series recorded while driving (large duration of time series, Table 2) into non-overlapped parts with a duration of 50 s. We opted for *cECG_2_* because it has a better recording quality compared to *cECG_1_* (higher amplitude, Table 1) and manual notation of useful segments available in Reference [28]. The total number of signals was 648, out of which 70% were used for the training set, and 30% for the test set. Data with a large amount of coarse artifacts, a moderate amount of coarse artifacts, and without coarse artifacts are uniformly arranged into a training set and a test set. For validation, we used only 10% of the training data because of the small size database. In the case of DDNN, the model has converged after several hundred epochs, so the number of observed epochs was set to 500. There are 2 classes in total, driving in the city and on the open road. We used a list of features: the value of the R peak, the HR (estimated as inversion of time intervals between adjacent R peaks), the BinEn of cECG time series, and pNN (percentage of successive normal cardiac interbeat intervals), that corresponds the percentage of RR intervals greater than 50 ms after reduction of artifacts (details are described in Reference [42]).

Classification performance is calculated according to the following expressions:(18)$$\mathrm{Accuracy}=\frac{TP+TN}{TP+FP+FN+TN},$$
(19)$$\mathrm{Sensitivity}=\frac{TP}{TP+FN},$$
(20)$$\mathrm{Specifity}=\frac{TN}{FP+TN},$$
(21)$$\mathrm{Positive}\;\mathrm{prediction}=\frac{TP}{TP+FP}\;,$$
(22)$$\mathrm{Negative}\;\mathrm{prediction}=\frac{TN}{TN+FN}.$$

The classification performance was tested in the context of quantifying the success of recognition of the driving location—open roads or city. In this context, *TP* denotes the number of cases correctly identified as driving in the city, *FP* denotes the number of cases incorrectly identified as driving in the city, *TN* denotes the number of cases correctly identified as driving in open roads (highway or polygon), and *FN* denotes the number of cases incorrectly identified as driving in open roads (highway or polygon).

In the context of quantifying the success of the artifact reduction by the proposed method, TP denotes the number of correctly identified useful segments (marked by medical experts), *TN* denotes the number correctly identified useless segments (segments without R peaks and categorized as artifacts according to experts), *FP* denotes the number of segments incorrectly identified as useless, and *FN* denotes the number of segments incorrectly identified as useful.

## 3. Results and Discussion

Figure 8a,b comparatively show the mean value of the percentage of eliminated time series by the proposed method, and the percentage of eliminated time series according to the notion of medical experts. These results are in excellent accordance for all three groups of *cECG_1_*, *cECG_2_*, and *cECG_3_* time series recorded while lying on the bed (Figure 8a). The high percentage of eliminated segments in time series while lying in bed is due to the movement of volunteers required by protocol. The high values of standard deviations shown in Figure 8 show a great variability of the amount of artifacts in the recorded signals. Figure 1a and Figure 2 show that the *cECG* time series contain different amounts of artifacts during driving. Besides, the controlled movement of the volunteers in an experiment driving car affects the electrodes in the upper part of the body more than it affects other electrodes [1].

The difference between the eliminated artifacts and medical expert’s opinion is larger for *cECG* time series recorded during driving, due to the slow-changing artifacts (Figure 8a). The elimination of slow-changing artifacts is complicated by the low intensity of the recorded signal while driving (Table 1), which makes the difference between a useful and a useless segment imperceptible (examples are marked in Figure 1c).

Figure 8b shows the high percentage of preserved R peaks marked by medical experts. Included in the analysis are all cECG time series that are labeled by medical experts. Unfortunately, notation of R peaks for *cECG_3_* time series recorded during driving are not available in Reference [28]. As a consequence of the fact that useful parts of the signal are comparable to the parts of the signal with slow-changing artifacts, slow-changing artifacts are partly survived, especially, while driving a car. Fortunately, the presence of slow-changing artifacts does not significantly affect the estimation of the mean value of the absolute amplitude cECG, which is confirmed in Figure 8c.

To check the presence of a significant difference between mean value of absolute amplitude of cECG after the elimination of the artifact by the proposed method and after the elimination of the artifact following the opinion of experts, we used a *t*-test for paired samples. The presence of statistical significance was observed between the group of raw recorded signals and signals after artifact elimination by the proposed method (Figure 8c, marked *), as well as to signals after artifact elimination, by the opinion of medical experts (Figure 8c, marked #). There is no statistical significance between the signals after the reduction of artifacts by the proposed method and following the opinion of experts.

Figure 9c,d show examples of the *cECG* time series after artifacts reduction by the proposed method in the car and on the bed, respectively.

Figure 10a,b show the percentage of preserved useful segments (segments with R peaks according to medical experts), the percentage of eliminated useless segments (segments without R peaks and categorized as artifacts according to experts), and overall accuracy (Equation (18)), depending on length SL. A high level (95–100%) of preservation of the useful segment of time series is achieved, as well as reduction of coarse artifacts (98–100%), while the percentage of eliminated R peaks is around 10% (except for *cECG_1_* recorded during driving with the lowest amplitude intensity) for a length of 0.5 s (Figure 10c–d) for all signal groups. Overall, accuracy is slightly lower for time series recorded while driving (Figure 10a), due to the impossibility of eliminating slow-changing artifacts as a consequence of the very low intensity of recorded time series (Table 1). In addition, it has been shown that slow-changing artifacts, which are not eliminated from the time series, do not significantly affect changes in statistical parameters (Figure 8c). The same problem was observed in Reference [1], where the impossibility of detecting QRS complex in segments that are comparable to noise.

To the best of our knowledge, there are two methods for reducing cECG signaling artifacts of mobile subjects (described in detail in References [1,18]). These methods estimate the duration of the interval between R peaks (detected by OSEA) in raw cECG and reject atypical values as artifacts. In Reference [18], the HR value that is larger than 120 beats/min and lower than 30 beats/min was treated as artifacts. The authors pointed out that, for real application, the procedure should be improved [18]. In Reference [1], the possibility of reliable estimation of HR during driving is investigated. The proposed method is based on QRS detection in raw cECG times series by OSEA software. It was noted that many false-positive QRS were detected by OSEA, as a consequence of the shaped pulse of cECG that is very similar to QRS complexes, so specific boundaries have been introduced [1].

A comparative analysis of the results of the proposed method the existing algorithms is not possible. Our method eliminates the artifacts before extracting the parameters, such as HR and QRS, and, more importantly, without predefined ranges. Thus, our method enables detection of potential cardiovascular pathology from the corrected signals, which is not possible in methods based on predefined ranges.

Statistical significance of entropy as a measurement of the complexity and unpredictability of time series [43] was observed between groups of ECG recorded with and without disturbing the driver while driving [44]. We test the possibility of applying binarized approximated entropy (BinEn), a method developed for entropy estimation on signals that do not require artifact elimination. We estimated BinEn for approximate entropy (BinApEn) and sample entropy (BinSampEn) for different groups of parameters. Statistical significance was observed in estimating BinApEn and BinSampEn between the raw signal and the signal after artifact removal by the proposed method only for parameters (*m* = 3, *r* = 1) for the *cECG_1_* group recorded in the car (time series with the smallest amplitude in Table 1). For other groups of signals with higher value of amplitudes, no statistical significance was noticed. The possibility for BinEn to distinguish between *cECG* recorded while driving in the city and on the open road (highway and polygon) after artifacts reduction is shown in Figure 11e.

The values of the threshold $TH_{1}\;$for *cECG* recorded during driving and while lying on the bed are shown in Figure 12. Introducing a constant threshold or range of values would not lead to an adequate result because the final value $TH_{1}$ depends on the statistical parameters of the time series. $TH_{1}\;$values are slightly lower for time series of lower intensity, recorded in the car, but there are isolated cases whose threshold value is measurable with the time series recorded while lying on the bed (higher intensity; see Table 1). In addition, eliminating coarse artifacts based on the amplitude value (e.g., >4 V is a coarse artifact) would not be a good solution because it is a value that would eliminate partly coarse artifacts in the case of a cECG recorded while driving, but, for cECG recorded during lying, it would eliminate R peaks (Figure 1a,b).

We use the KNN technique [38] and DDNN technique [39] to classify driving in city conditions and in the open road (highway or on the proving ground). To investigate the sensitivity of KNN and DDNN on artifacts, we compare results on raw cECG and cECG after artifacts reduction. The list of features consists of the value of the R peak, the HR, and the BinEn of cECG time series. Table 4 shows the impact of artifacts on the accuracy of machine learning techniques. Results of the KNN technique [38] show that 23% accuracy has been improved for *cECG* time series after artifact elimination (in Table 4, noted as KNN^2^) and 39% for DDNN techniques [39] (in Table 4, noted as DDNN^2^). In addition, 55% growth was achieved for sensitivity and for positive prediction of 59%, while, for the DDNN, it was even growth for 73.47%, 63.18%, respectively. The accuracy of KNN classifications has increased after the expansion of the feature list with pNN50, a method which requires the reduction of artifacts (in Table 4, noted as KNN^3^), but, for DDNN, it was a slight increase of 0.51%. In addition, we can note the increase of all classification performance for KNN^3^ in comparison to KNN^2^. DDNN is more sensitive to the presence of artifacts compared to KNN, but it is also slightly accurate compared to KNN.

## 4. Conclusions

The main contribution of this paper is the development of a method for the reduction of artifacts in cECG signals. The detailed analysis of cECG signals publicly available on [28] reveals that medical experts have concluded that about 30% of the signals represent noise, as well as that there are only a few cECG time series without coarse artifacts or with a moderate amount of artifacts. Such domineering presence of artifacts is not aligned with the theoretical requirements according to which the signals should be stationary and without the artifacts. Besides, transferring the recorded cECG signals to the cloud, without prior reduction of the artifacts, would significantly increase the amount of transmitted traffic, while data processing in the cloud would not be reliable. The proposed method can be applied online. The first 50 s of cECG provide statistical parameters of $F_{D}$ and the corresponding thresholds. Each subsequent recorded cECG segment affects the expansion of the$\;F_{D}$ series of fluctuation values and updates the threshold value. To strengthen the motivation for our work, we have also shown that, even in a small size database, it is noticeable that the presence of artifacts weakens the performance of the classification of KNN and DDNN machine learning techniques, which differentiate urban and other driving conditions. Unfortunately, a small database can affect reliability in estimating the accuracy of classification, especially of DDNN techniques. An alternative to reducing artifacts is to develop methods that are resistant to the presence of artifacts. As an example, we have shown binarized entropy that operates on binary differential coded raw signals and yields good entropy estimates. The pseudocode is given for the easier implementation of the algorithm, and the code is available on request.

## Figures and Tables

**Figure 1 entropy-24-00013-f001:**
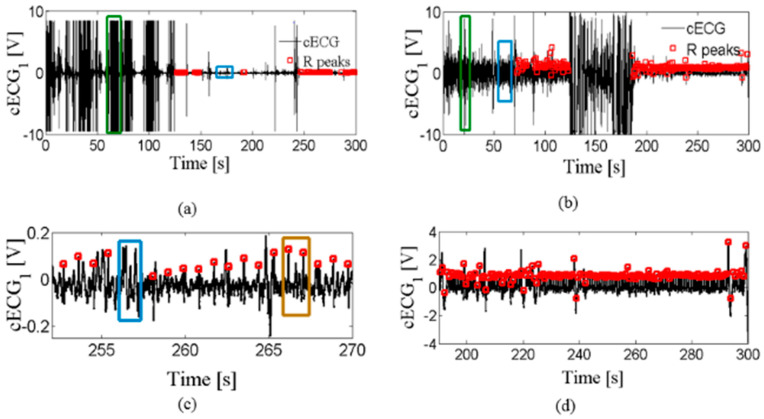
An example of a cECG time series recorded (**a**) while driving in car, (**b**) while lying on the bed, (**c**) enlarged part of useful segments during driving, and (**d**) enlarged part of useful segments while lying. Red rectangles indicate R peaks marked by medical experts; an example of coarse artifacts is marked by the green rectangular border in (**a**,**b**); examples of slow-changing artifacts are marked by the blue rectangular border in (**a**–**c**). An example of the useful signal segment that is similar to the useless segment is marked by the brown rectangular border in (**c**). Slow-changing artifacts were determined by checking the overall accuracy in eliminating useless signal segments.

**Figure 2 entropy-24-00013-f002:**
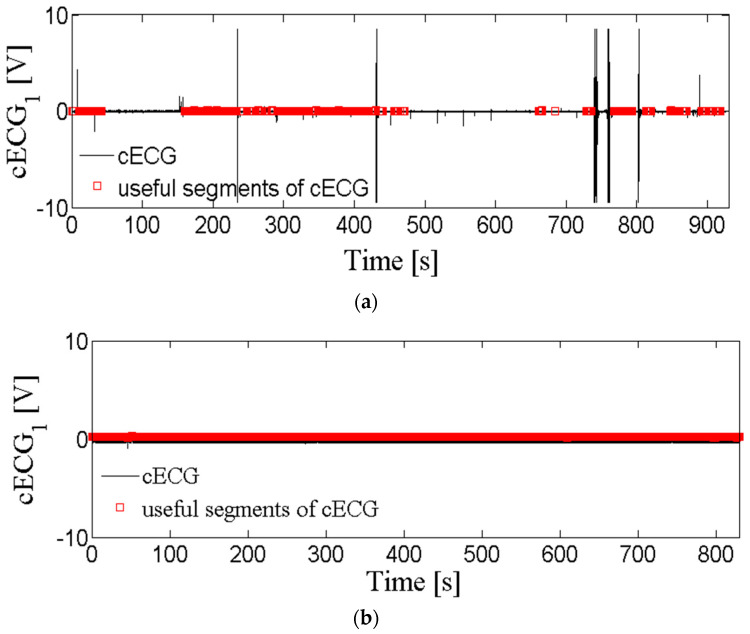
cECG time series recorded during driving: (**a**) *cECG_1_* with a moderate amount of coarse artifacts; (**b**) *cECG_1_* without coarse artifacts. Red marks useful segments according to medical experts.

**Figure 3 entropy-24-00013-f003:**
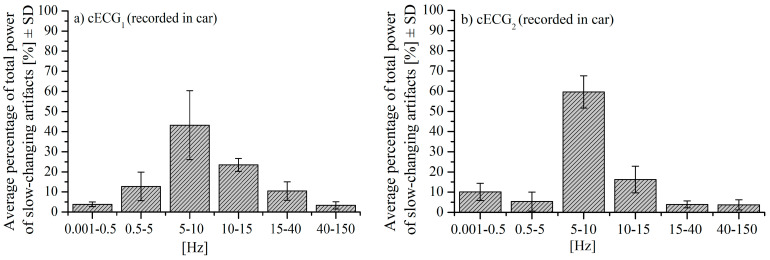
Mean value of percentage of total power of slow-changing artifacts over frequency bands ±*SD*: (**a**) *cECG**_1_* during driving; (**b**) *cECG**_2_* during driving.

**Figure 4 entropy-24-00013-f004:**
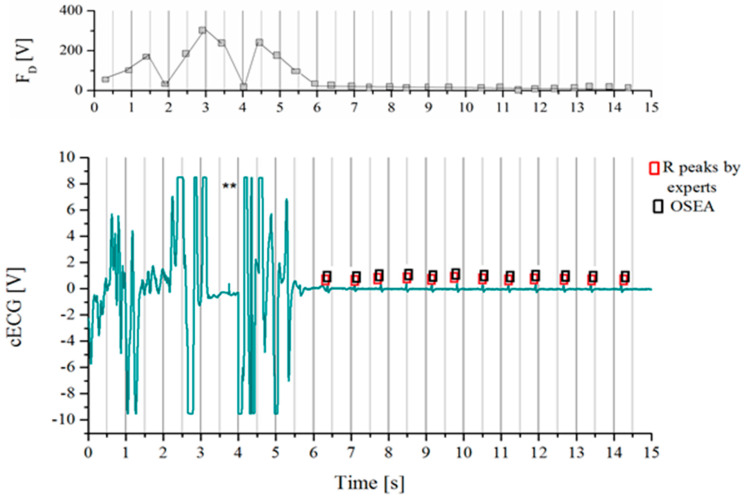
Example of extracted part of raw *cECG* time series recorded while driving. Red rectangles indicate *R* peaks of *cECG* signals marked by medical experts, black rectangles indicate peaks detected using OSEA software. Gray vertical lines indicate segments for which the value of the estimated fluctuation *F_D_* is shown in the upper panel. Asterisks (**) indicate characteristic segments that appear next to segments with coarse artifacts.

**Figure 5 entropy-24-00013-f005:**
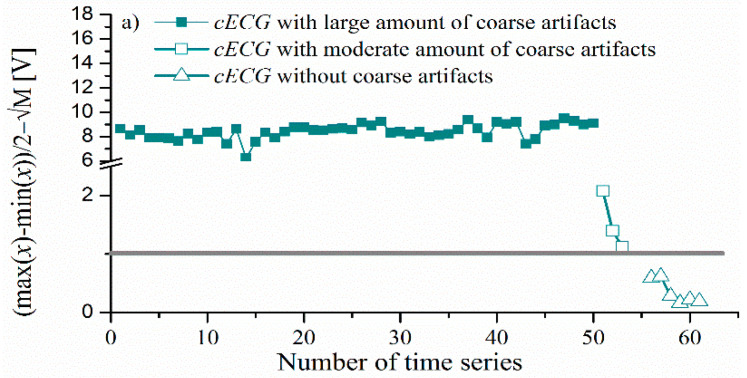
Difference between half of the difference maximal and minimum amplitude of the time series, and the square root of the second moment (**a**) *cECG_1_* and *cECG_2_* time series recorded while driving, (**b**) *cECG_1_*, *cECG_2_*, and *cECG_3_* time series recorded while lying in bed.

**Figure 6 entropy-24-00013-f006:**
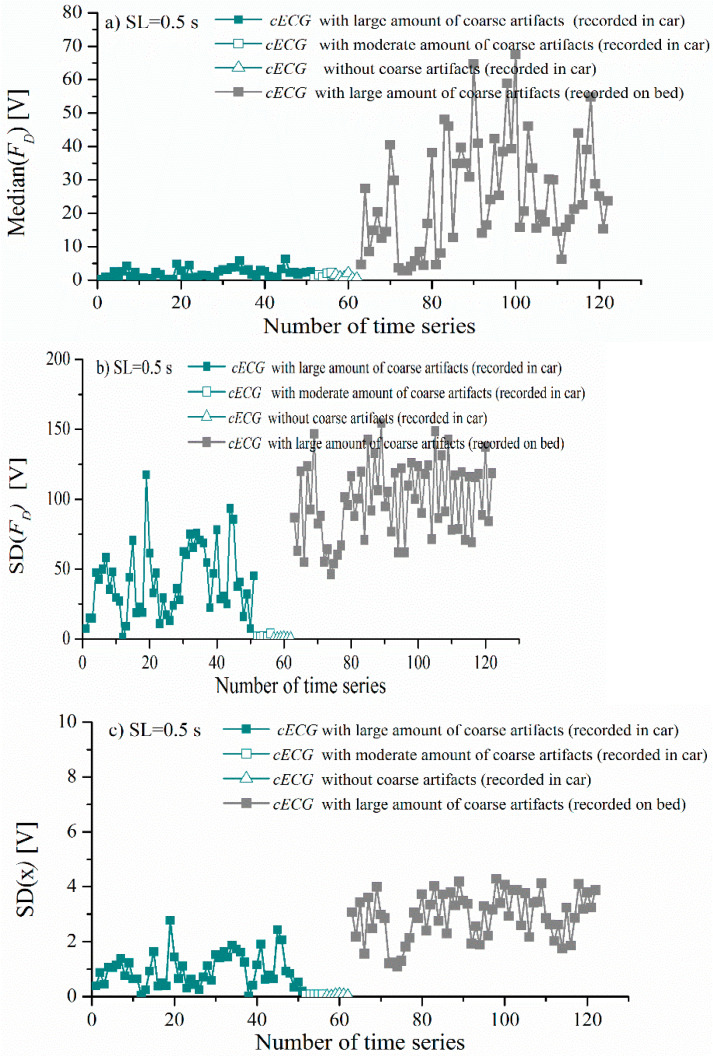
Statistical parameters of the recorded *cECG* time series for different recording conditions: driving car, lying in bed. (**a**) Median value of *F_D_*, (**b**) standard deviation *SD*(*F_D_*), and (**c**) standard deviation of *cECG* time series.

**Figure 7 entropy-24-00013-f007:**
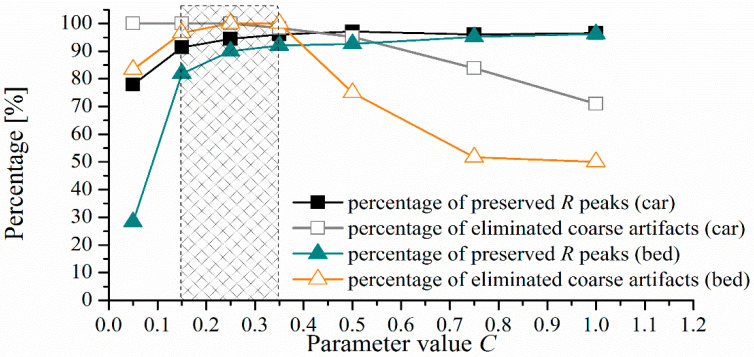
Influence of empirical parameter C on percentage of preserved R peaks marked by medical experts and percentage of eliminated coarse artifacts for *SL* = 0.5 s.

**Figure 8 entropy-24-00013-f008:**
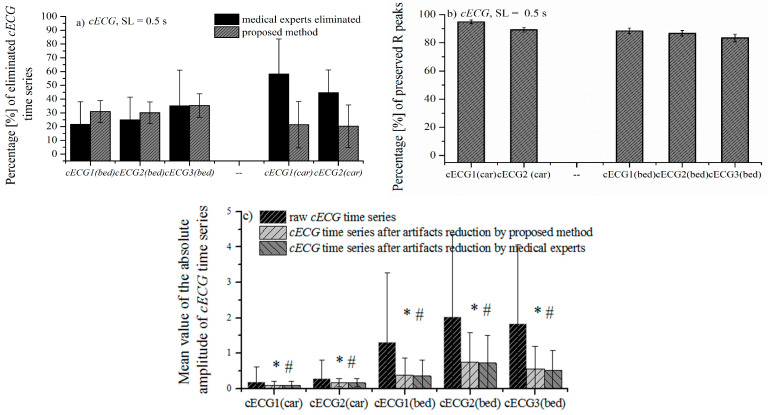
Comparative presentation of the mean value ± SD percentage of eliminated time series by the proposed method and by the opinion of experts: (**a**) *cECG* time series recorded while lying on the bed, (**b**) *cECG* time series recorded while driving and (**c**) mean values of absolute amplitudes of *cECG* time series ± SD. Statistical significance is observed between the raw *cECG* and *cECG* after elimination of the artifacts by the proposed method (marked *), as well as the *cECG* after elimination, according to the experts (marked #). Statistical significance does not exist between the group of signals after elimination of the artifact by the proposed method and in the opinion of experts. We used a *t*-test for paired samples, with significance levels *p* < 0.01 for all compared groups.

**Figure 9 entropy-24-00013-f009:**
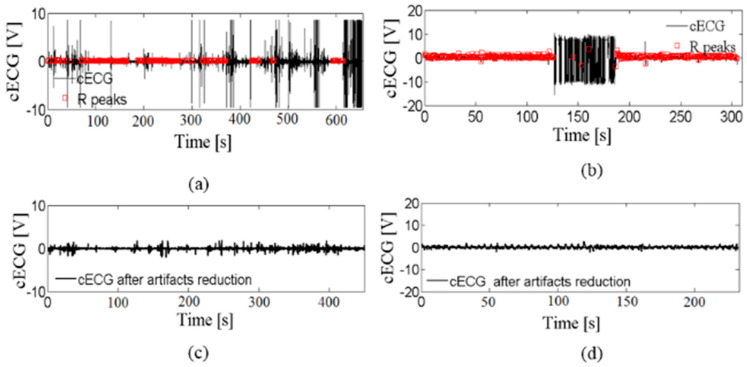
(**a**) Raw *cECG_1_* time series recorded while driving, (**b**) *cECG_3_* time series recorded while lying on the bed, (**c**) *cECG_1_* time series after elimination of artifacts by the proposed method, and (**d**) *cECG_3_* time series after elimination of artifacts by the proposed method.

**Figure 10 entropy-24-00013-f010:**
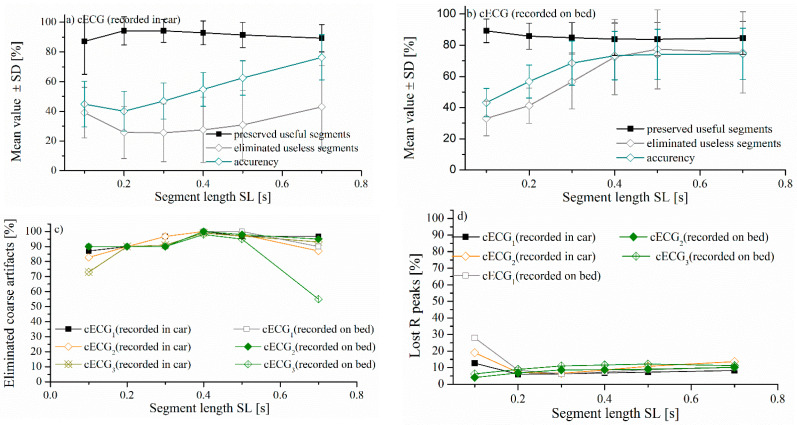
Demonstration of the success of eliminating artifacts and preserving the useful part of the signal by the proposed method depending on SL. (**a**) *cECG_1_*, *cECG_2_* recorded in the car, (**b**) *cECG_1_*, *cECG_2_*, and *cECG_3_* while lying on the bed, (**c**) the percentage of time series in which the coarse artifacts are fully eliminated, depending on the SL, and (**d**) the percentage of lost R peaks after artifacts reduction, depending on the SL.

**Figure 11 entropy-24-00013-f011:**
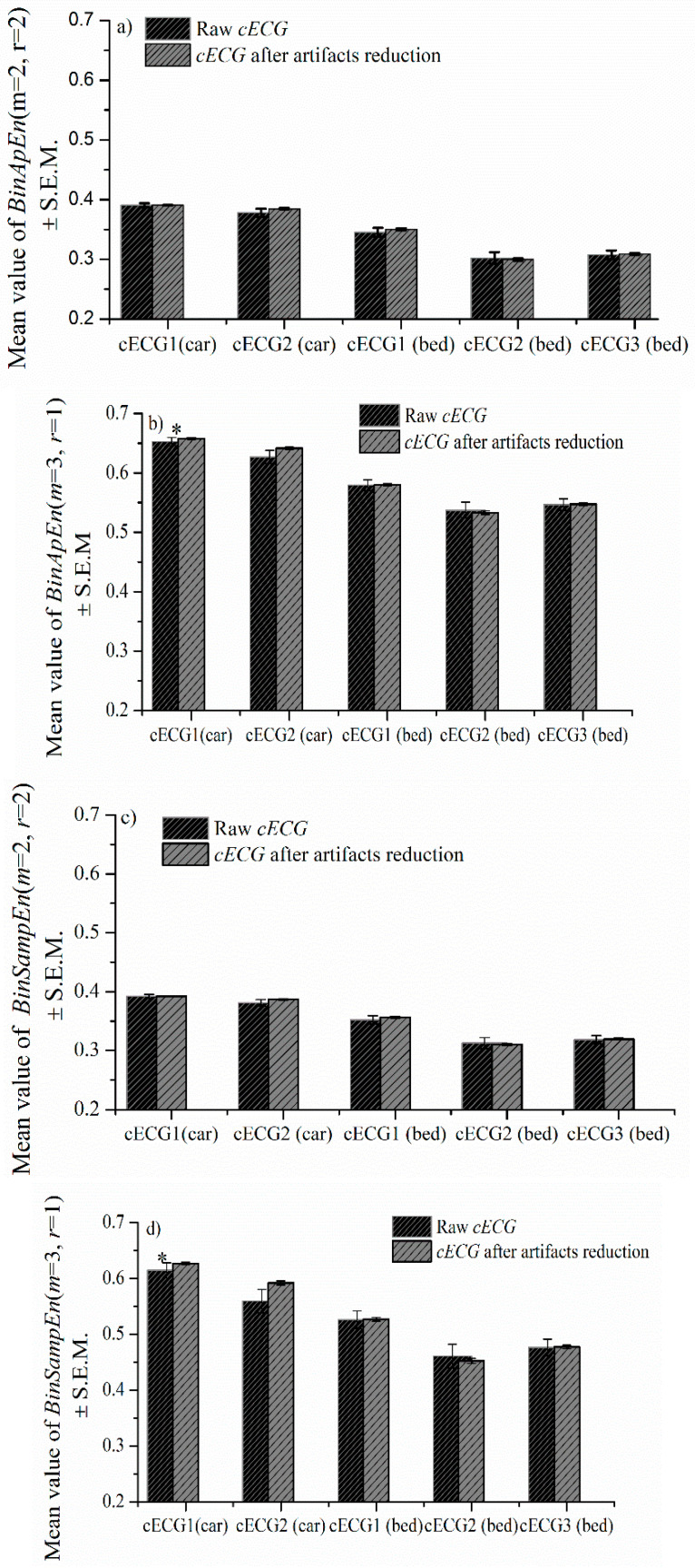
*BinApEn* and *BinSampEn* for *cECG* of all observed groups. (**a**) Mean value of *BinApEn* (*m* = 2, *r* = 2) ± SD, (**b**) mean value of *BinApEn* (*m* = 3, *r* = 1) ± SD, (**c**) *BinSampEn* (*m* = 2, *r* = 2) ± SD, (**d**) *BinSampEn* (*m* = 3, *r* = 1) ± SD and (**e**) mean value of *BinApEn* ± SD and *BinSampEn* ± SD for (*m* = 2, *r* = 2) and (*m* = 3, *r* = 1) of the cECG recorded during driving in the city and in the open road. Statistical significance between BinEn of cECG after elimination of the artifacts during driving in the city and the open road is marked by *.

**Figure 12 entropy-24-00013-f012:**
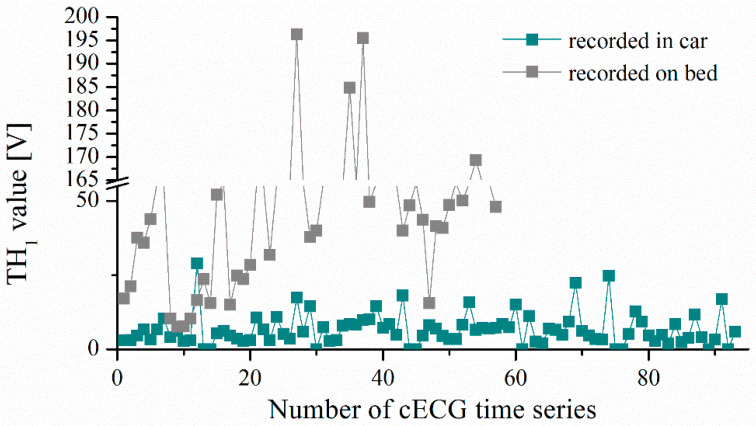
Threshold value of *TH*_1_ for *cECG* of all observed groups.

**Table 1 entropy-24-00013-t001:** Absolute amplitude of *cECG* [V], expressed as a mean ± standard deviation *SD*.

	Number of Measurements in Each Group	*cECG* * _1_ *	*cECG* * _2_ *	*cECG* * _3_ *	Reference Signal
CAR	31	0.19 ± 0.67 [V]	0.29 ± 0.89 [V]	0.21 ± 0.69 [V]	0.81 ± 0.93 [V]
BED	20	1.18 ± 2.22 [V]	1.85 ± 2.64 [V]	1.66 ± 2.58 [V]	0.37 ± 0.52 [V]

**Table 2 entropy-24-00013-t002:** Duration of extracted ECG segments [s], expressed as a mean ± standard deviation *SD*.

	Number of Measurements in Each Group	*cECG* * _1_ *	*cECG* * _2_ *	*cECG* * _3_ *	Reference Signal
CAR	31	1556.93 ± 2166.13 [s]	1556.93 ± 2166.13 [s]	1556.93 ± 2166.13 [s]	1556.93 ± 2166.13 [s]
BED	20	312.45 ± 10.54 [s]	312.45 ± 10.54 [s]	312.45 ± 10.54 [s]	312.45 ± 10.54 [s]

**Table 3 entropy-24-00013-t003:** Number of cECG per observed groups during driving.

	*cECG* * _1_ *	*cECG* * _2_ *
cECG with large amount of artifacts	26	25
cECG with moderate amount of artifacts	2	3
cECG without artifacts	3	3

**Table 4 entropy-24-00013-t004:** Classification performance [%].

ML Technique	Accuracy	Sensitivity	Specificity	Positive Prediction	Negative Prediction
KNN^1^	65.64	14.28	82.87	21.87	74.23
KNN^2^	88.21	69.40	94.52	80.95	90.20
KNN^3^	92.68	77.55	97.95	92.68	92.86
DDNN^1^	53.33	20.41	64,38	16.13	70.68
DDNN^2^	92.31	93.88	91.78	79.31	97.81
DDNN^3^	92.82	95.83	91.84	79.31	98.54

KNN^1^ for raw *cECG*. KNN^2^ for *cECG* after reduction artifacts. KNN^3^ for *cECG* after reduction artifacts with extended features with pNN50. DDNN^1^ for raw *cECG**.*** DDNN^2^ for *cECG* after reduction artifacts. DDNN^3^ for *cECG* after reduction artifacts with extended features with pNN50.

## Data Availability

Data is contained within the article. Datasets are available on https://www.medit.hia.rwth-aachen.de/publikationen/unovis/ (accessed on 26 May 2011).

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
