# Peer review of "Reduction of Artifacts in Capacitive Electrocardiogram Signals of Driving Subjects"

_entropy, 2021, doi:10.3390/e24010013_

Round 1
Reviewer 1 Report
With the rise of smart cars and e-health services, there is an increased demand for understanding the health conditions of drivers. This paper specifically attempts to improve the accuracy of detecting certain EEG signals, and therefore leading to a more accurate depiction of drivers’ health.
While the methods are clearly described, perhaps there could be an explanation as to why the highway and polygon were tested (What is the difference between these two conditions? Is this something that has already been established in the literature?). The procedure for artifact reduction is described thoroughly by explaining the fluctuation estimation and providing a clear explanation on the formulas used for establishing a threshold. The limitations are clearly discussed and there are solutions provided to account for these issues. There is also a concise summary at the end of the pseudo code which thoroughly describes the algorithm.
The authors describe the materials and methods used well, however, there could be some improvements of why these methods were chosen to provide more context. The results are clearly depicted through tables and figures that help to visualize the results. This is novel work that attempts to improve current methods of analyzing cECG data.
Reviewer 2 Report
This study presents a method for reduction of artifacts in ECG signals, specifically induced while driving and lying on the bed. Although the method could be promising, the quality of the overall presentation is not satisfactory for publication. The most important drawback are the Methods - the terms and methodological steps are not fluently presented and a total comprehension is a mismatch. Besides, methods miss a lot of analyses, which are reported in Results.
The article needs considerable changes following the disclosed detailed revision remarks in all major sections Abstract, Introduction, Methods and Results. If the authors decide to resubmit a new version of their paper, then I will strongly advice to provide a point-by-point answer to EACH revision remark (to the reviewer and highlighted text in the main manuscript file). Do not group answers to several remarks or dismiss. This would surely lead to further negative decision for rejection.
Major revision remarks:
- Abstract: The term “slow-changing artifacts” is a main characteristic of the target signals, however, it is out of sense without specification of the artifacts’ frequency band. It is questionable if the bandwidth of the “slow-changing artifacts” overlap with the diagnostic (0.05-150Hz) or monitoring (0.67-40Hz) frequency band of ECG signals. It is important to define the complexity of the task if both bandwidths significantly overlap. This numerical information should be given as a statistical measure on the database in the Abstract’s “virtual” sub-section Results. Besides, it would be important to report the statistical value of the signal-to-noise ratio (SNR) in your artifact database so that the reader is convinced about the database content. These important issues should be also included in the main text.
- Abstract: Ln 15-20: I note a great discrepancy in the Abstract, presenting Methods in “virtual” section Results. This is an indicator for a wrong structure of the abstract, which should be formatted according to the journal’ instructions for authors (https://www.mdpi.com/journal/entropy/instructions): “The abstract should be a single paragraph and should follow the style of structured abstracts, but without headings: 1) Background: Place the question addressed in a broad context and highlight the purpose of the study; 2) Methods: Describe briefly the main methods or treatments applied. Include any relevant preregistration numbers, and species and strains of any animals used. 3) Results: Summarize the article's main findings; and 4) Conclusion: Indicate the main conclusions or interpretations.”. In fact, I miss sufficient information in the Abstract’s sub-sections: Methods, Results (most important numerical findings), Conclusions (clinical implications based on the results). The abstract should be fully rewritten in major part.
- Abstract: Write ONLY the essential part of the study without using too explanatory, trivial details, such as “a machine learning technique K Nearest Neighbor”, i.e. “machine learning technique” is a well known fact that should be skipped. Indeed important details of methods and results are missing and the comprehension of the overall abstract is very low. In overall the presented abstract is NOT a sufficient representation of the article.
- Keywords: They do not represent the main keywords related to this study. The authors MUST mention global keywords and definitions for index search common within the subject discipline, but not a free-text. For example, the term “signal fluctuation” does not represent the truthful term common for ECG signal processing, namely :”ECG drift”; “reduction of coarse and slow-changing artifacts” does not represent the common term “ECG filtering” or (the name of specific filtering methods), etc. Think carefully about the index terms – they should be extracted from “Index of terms” or represent specific definitions in the subject area.
- Technical remark: remove all “Italic text”, as cloud, ECG, Heart Rate-HR, Active Noise Cancellation, Empirical mode decomposition, etc. all over the text. All these terms are common abbreviations or normal part of the text, and there aren’t any reasons nor guidelines to highlight them as Italic text. This is rather confusing. Concerning the definition in Ln 38 “(Heart Rate-HR)”, turn attention that the first-time abbreviation should be in brackets but not the whole definition, i.e. “…. for heart rate monitoring (HR).”
- Ln 75: “open-source OSEA” -> The term OSEA is used without a definition. Check if all abbreviations are correctly defined on their first use over the whole text.
- Ln 103-121: The aims of the study are not properly defined. They should represent a brief definition of the novel methodologies applied for solving of the problems as exposed in Introduction. The aims should be a formal definition of ideas, without going into methodological details as they are not comprehensive at this part of the article. Aims should finish with a brief justification of the EXPECTED results, benefits and comment on clinical implications. The overall text is normally fitted into one paragraph with up to 5-10 lines.
- Ln 124-132 – Missing flow diagram that explains the relations between all modules. If not relevant, normally this introductory section is skipped because it is quite confusing to understand methodological details, which are still not defined.
- The name of the section “2.1. cECG Time Series Acquisition” is NOT representative to the standard definition “Materials” or “Database”, as expected in the first sub-section of “2. Materials and Methods”. Generally, avoid the use of too explanatory name of sections.
- Ln 134-154: This section does NOT properly identify the strict protocol followed for data acquisition. According to the declaration of Helsinki, all studies involving humans should also be supported by a decision of an Ethical committee. I strongly insist that all details related to the registering of the study, Ethical committee decision (place/date/number), inclusion/exclusion criteria of subjects, as well as the pre-defined protocol for data acquisition is systematically disclosed.
- Ln 144: “the equipment of Mr. Bsamp, Mr. Tec (Guger technology)” -> The name of the ECG device (the name of the manufacturer) should be correctly referenced.
- Ln 145: “The sampling frequency was 1000 Hz, except for the two signals with a sampling frequency of 200 Hz.” -> Indefinite explanation of the ECG leads. This is very important setting of the study, so that the article must strictly identify the acquitted ECG leads (the position of the electrodes at the point of contacts with the body, namely Electrode 1, Electrode 2, Electrode 3, Reference ECG lead), the amplitude and the frequency resolution for each ECG lead.
- Table 1, Table 2: The notations cECG1, cECG2, cECG3 are not identified in the text for the database with lying persons. It is quite confusing because 12 built-in electrodes in the bed are disclosed in Ln 147. In overall, the most important reason to disclose Table 1 and Table 2 is not justified in the main text.
- Table 1, Table 2: “cECG signal (V)±??”, “length of time series (samples)±??” -> Properly identify the content of both tables. Seems to be incorrect and incomprehensive reference to the disclosed values. Besides, the term “length of time series” is also incorrect but should reveal the “duration of extracted ECG segments”.
- Table 1, Table 2: Identify the measurement units of all digital values. In Table 2, the numbers seem very large and incomprehensive. If these are number of samples, then you need to recalculate to the universal SI unit of seconds. Never use a system-specific units as “number of samples”, which depend on the sampling rate. These numerical values are mostly incomprehensive to the readers and cannot be directly linked to other studies or methods.
- Table 1, Table 2: Identify the number of records at different parts of the study protocol, as defined in the text.
- Ln 160 “Table I shows”, Ln 162 “Table II” -> I could not identify any of both Table I and Table II but only Table 1 and Table 2. Strictly refer the exact numbering of tables and figures all over the text.
- Ln 168-169: “Red markers point the annotated R peaks according to medical experts.” -> In fact, the authors need to justify how exactly R-peaks are used, i.e. for heart rate measurement, considering that the acquired cECG during driving is NOT used for the standard cardiologist-level diagnostic purposes. In this context, the authors need to identify how the HR measurements are exactly managed in cases of artifacts (in conditions with unreliable QRS detection, where most likely are false positive or false negative QRS detection errors), i.e. how the short-term and long-term HR is being measured.
- Figure 1: The small rectangles with embedded signals are unreadable and incomprehensive! If the authors want to present something important, then these rectangles are necessary to be enlarged, depicted in additional subplots and most important findings explained.
- Ln 173-174: Use of terms without definitions. The overall statement is completely incomprehensive.
- Ln 181: It is not clear how the cumulative sum in equation (1) can be divided in non-overlapping segments. It is physically inapplicable.
- Ln 183-184: “perform the time series splitting procedure from the other end to analyze all parts of the 183 time series” -> Clarify the statement.
- Ln 186: “will” -> Never use future verb tense in scientific articles unless they refer to some future plans in Discussion. This is not the case, therefore, it indicates about a wrong statement. Correct all such discrepancies within the text.
- Ln 187-188: “In shorter time series, the division should be repeated starting from the last sample [31].” -> Clarify the statement.
- Ln 194: “method is linear (v=1)” -> The replacement (v=1) is not identified in any formula, therefore, it is not clear how to interpret.
- “implemented linear approximation as well” -> It is not identified in any formula the other type of approximation, which is suggested by the statement “as well”.
- Ln 196 “The squared fluctuation of one segment” -> check the term “squared fluctuation”.
- In overall, the term “Fs” in equation (3) is not sufficiently justified. Generally in most studies, Fs denotes sampling frequency, therefore, it is not clear if the authors do not apply resampling of the data. Clarify the meaning and interpretation of Fs in Figure 2. What is the range of Fs values and how to interpret low and high values.
- Ln 212-213: “Figure 2 shows that R peaks detected by OSEA software are in accordance with experts' opinions in useful part” -> Absolutely incomprehensive. There are not disclosed any guidelines on how to identify the “useful part”. It is strange that only 2 valid QRS complexes are detected within such long recording. They CANNOT be used for any comprehensive measurements. Therefore, systematically justify the meaning of the performed ECG measurements as a result of the proposed interpolation.
- Figure 2: The x-axis should be presented in universal units of seconds but not in samples.
- Ln 214-215: “Unfortunately, the amplitudes of the useful parts are comparable to the parts of the signal with slow-changing artifacts in which the cECG was not detected.” -> Clarify the statement.
- Ln 218-220: “From Figure 2 can also be seen that there is a considerable deviation between the useful part of the time series marked by medical experts and the parts in which the artifacts are dominant.” -> clarify the statement. As defined in some comment above, the useful and useless parts of the signals are not identified, so the reader cannot comprehensively follow the explanations and ideas. The authors seem to be totally confused about how to explain the principle in an accessible and clear way. The text of the section should be rewritten in the most part.
- Ln 225: “To reduce the presence of coarse and slow-changing artifacts” -> The definition of the terms “slow” and “coarse” should be first explained together with criteria for their separation in the frequency band. Also comparison with the diagnostic frequency band of ECG signals is not disclosed, clearly identifying the level of expected frequency overlapping.
- Ln 227-230: The thresholds and terms MUST be defined upon their identification with respective formula. The overall explanation is totally confused and incomprehensive!
- Ln 231-234: The overall explanation is confusing and must be totally rewritten.
- Ln 235-236: Justify the statement. Such conclusion can be derived only after analysis of some results. Such are missing and cannot be disclosed in Methods.
- Figures 3 and 4 are used without definition of the terms in Methods. I strongly insist that all depicted terms are first presented with formula in Methods. Furthermore, the font is very small and unreadable. Missing measurement units of y-axis. It is absolutely incomprehensive how to interpret Median (F), SD(F) and SD(x), cECG(CAR), cECG(BED) as those terms are NOT defined and explained in Methods with and without coarse artifacts. Who and how identified the presence of artifacts? Small note that the terms (CAR) and (BED) are not corresponding to the definitions of the database in Tables 1 and 2.
- The overall reference to Figures 3 and 4 in Ln 237-248 is confusing. It must be rewritten in most part.
- Ln 254-262: Incomprehensive justification of ideas. The reference to Figure 4 is a total mess. The definition and use of TH1 is questionable and the particular use of TH1 in Figure 4 is not observable.
- Equation (4) uses a lot of terms without a definition. Make sure that each single argument in this long equation is well justified in the paragraph just before or just after equation (4) and that it is properly inked to either Figure 4 or Figure 5.
- Figure 5 is not linked to the parameter C -> not seen.
- Ln 279: “segment length SL = 500” -> What is this? The segment length should be given in units of seconds but not in samples. It is questionable if this method could work with another sampling rate or is fixed to exactly 500 Hz. If the latter, then the method is not generally usable and does not deserve to be published. A more generalizable version is required.
- The definition of the thresholds TH2 and TH3 is not clearly explained in page 9.
- Ln 315-348: The need and steps of the algorithm are not clearly explained in the text. In overall, methods do not systematically present all steps of the algorithm. I strongly recommend the authors to present their Methods, sequentially explaining step-by-step all details of their algorithm.
- Ln 352: “The subjective test was a visual inspection to confirm the absence of coarse artifacts” -> Estimation of results based on visual subjective inspection is quite confusing statement!!! The process for the study design should be disclosed in Methods. Only after that, the evaluated numerical values can be disclosed in Results.
- Figure 6: All figures seem to present statistical measurements and between group comparisons. However, Methods do NOT disclose any Statistical analysis of data, therefore, the reader is blinded how these statistical evaluations are performed and how to interpret the data. The p-values of statistical significance must be disclosed in results in all between-group comparisons. Methods should be amended accordingly.
- Ln 377-370: Clarify the statement,
- Ln 434-447: Includes Methodological details, which are NOT disclosed in Methods! Furthermore, some methodological settings (m= 3, r= 1) are defined, however, they are incomprehensive and meaningless. Keep in mind that every report of results should be linked to some Methodological section. Methods should be amended accordingly.
- Ln 455-461: These observations are hardly to be identified in Figure10. Strongly recommend to reformat the figure so that appropriate comparisons can be done. The ratio of the difference of difference car vs. bed should be identified for each case. Furthermore, the reference to Figure 1 is incomprehensive.
- Ln 464: “Results of the KNN technique” -> What is this? Such technique was NOT identified in Methods. Methods should be amended accordingly to explain all details and justify the application of KNN.
- Ln 466-467: “Data were divided into a training set (70%) and test set (30%), features are normalized.” -> This definition of the database should be part of methods. All features should be disclosed in methods. Here, absolutely incomprehensive!
- Table 3: Classification performance: It is questionable what kind of classification is dobe ny KNN. The formula for computation of all measures of performance (Accuracy, Sensitivity, Specificity, Positive Prediction, Negative Prediction) should be disclosed in Methods. Otherwise, the results are NOT interpretable.
- Ln 468-476: Incomprehensive description of Results.
- Ln 486-488: Incomprehensive definition of the main contributions with a reference to [37]. Is it true that the methods are not novel and previously developed in [37]?
Reviewer 3 Report
Please address the following comments before resubmitting.
Comment #1: Have you expiremented using deep learning algorithms (e.g. CNN) to classify the cECG signals? They are usually more robust than classic machine learning techniques.
Comment #2: In lines 130-132 you are referring to three groups of measurements, but mention only the two of them.
Comment #3: Why have you chosen the median point C = 0.25 as the threshold value? In Figure 5 it seems that values closer to the right edge of the box (0.25 < C < 0.35) also give good (or even better?) results regarding the percentage of preserved R peaks and eliminated coarse artifacts.
Comment #4: As you mention, there was no statistical significant difference in binary entropy measurements between raw and denoised signals except for the cECG1 group ( car ). Why do you argue that it is a good alternative for reducing artifacts in eECG signals?
Comment #5: Please revise the manuscript for grammar and spelling mistakes.
Round 2
Reviewer 2 Report
After the first revision review, the authors have improved their manuscript. However, there are still a number of important issues which should be clarified and considered before taking the final decision for publication of this paper.
Major revision remarks:
- Title: Abbreviations are NOT applicable in titles. All abbreviations should be used after their explanation in the text. Use a comprehensive title.
- Abstract: “Spectral bands of slowly changing artifacts and cECG signals overlap, the most in the band from 0.5 to 15 Hz, preventing classical filtering.” -> This statement presents the database (part of methods) and can be used for justification of methods. This statement is absolutely irrelevant for conclusions, where results should be commented in respect of their clinical implications.
- Ln 77: “open-source EKG analysis” -> Use the English term ECG.
- Ln 90: “Active Noise Cancellation (ANC)” -> The term is not a name and should be written with small letters. Check all such discrepancies all over the text.
- Ln 148: “but the amplitude values are very low, especially in driving conditions” -> Clarify the statement. I suggest a duplicated comment in Ln 155-159. This is confusing.
- Figure 1: Low quality and incorrect scale dimensions going out of page. This figure should be correctly positioned and scaled in a readable fashion.
- Figure 1: What is the difference between “Red rectangles” and “Red rectangular points”? I would recommend using different colors for indication of different things. Use more convenient explanations.
- Figure 2: The x,y size ratio of the figure is changed than original, so that the overall figure is distorted and not suitable for publication. Furthermore, do not duplicate the text in the caption and the figure title. Both are now different and in fact the one above the figure appears with wrong English wording.
- Ln 172-182: The overall text presents Methods and is NOT relevant in section “Materials” dedicated for description of the database. The overall paragraph is quite confusing in this section of the Manuscript.
- Ln 188-191: Clarify the statement. I suggest that this paragraph concerns Methods and is out of sense upon description of the database.
- Ln 260-274 and Figure 4 are relevant for section “2.1. Materials” because their purpose is to present statistics of the database itself. It is out of sense to be presented in Methods.
- Section “2.2. Motivation for” -> This phrase is irrelevant for the section’s title. I would recommend just typing: “2.2. Reduction of artifacts based on fluctuation in cECG time series”
- Ln 122-124: Justify the need for forward-backward ECG filtering but not applying a forward second order filter. The first approach is not applicable to real-time signal processing and the need for its application should be explained.
- Equation (2) contains two equations. This is confusing and the formatting should be corrected.
- Generally, the notations in Figure 3 should correspond to the basic notations in Figure 1 concerning the database. The correspondence between the QRS detection marks in Figure 1 and those detected manually and automatically in Figure 2 is questionable. At least use the same marks and explain the differences. In general, it is not applicable to duplicate the same information in different sections, therefore, the authors should consider merging of information.
- Ln 302-311: This section describes the content of the database and its split into three parts. It is relevant for section “2.1. Materials” but such description of the database is quite confusing in section Methods. Generally, the number of cases in each part should be reported in section “2.1. Materials” so that section Results corresponds to the exact number of cases as reported in Materials.
- Ln 473: Do not duplicate abbreviations in the titles and in the text: “2.4. K Nearest Neighbor (kNN) and Deep Dense Neural Network (DDNN)”. Better define them in the main text.
- Ln 476-477: The decision for setting K=5 should be justified. What are the main expected clusters? What type of optimization criteria and assumptions are applied? The details for classification methods should be provided in a comprehensive manner.
- Ln 478-481: The architecture of the network should be presented in enough detail to justify the input and the output, as well as the hidden layers definitions. Justification and optimization of the network hyperparameters should be comprehensively presented. The details for classification methods should be provided in a comprehensive manner.
- Ln 482-483: “Data were divided into a training set (70%) and test set (30%), features are normalized.” -> This description is crucial for the performance of any algorithm, therefore, it needs more attention. At least, the authors must referrer the table, which indicates the number of records used for calculation of the 100% database. Furthermore, they need to explain how they use the split of the database into three parts as defined in the previous comment (N16).
- Ln 482-495: Should be presented in section “2.5 Statistical analysis” as those represent definitions of the basic statistical metrics of performance.
- Ln 495: The term TP, FN, TN, FP should be explained in the context of the solved classification task – what exactly they mean in your study? For the QRS detection task, the term TN means nothing, which is confusing in your case, so that you need to explicitly clarify.
- Ln 506-507: This paragraph is meaningless.
- Ln 508-510: The decision for “deletion” of time-series is NOT explained in Methods. The exact term “delete” or “deletion” should be strictly explained and justified in Methods before reporting in Results. This is quite CONFUSING, considering that the term “artefact reduction” is mentioned in Methods and part of the figures.
- Figure 9: The size ratio of all subplots is different, and as a result the figure looks ugly and unsuitable for publication. Al subplots should be resized with the same ratio, the text should be with the same size. Furthermore, the figures should be zoomed to their meaningful length, i.e. the last part of the figures is lost for no signal.
- Figure 10: The same problem with the figure size as noted above. All subplots should be resized under equal conditions, so that the text and traces should look in the same size and scale.
- Figure 10: The metrics in the figure Legends (“true positive”, “true negative”, accurency”) is NOT corresponding to the definition in Methods (equations 17-21). A strict correspondence of the terms in ALL Figures, Tables, Methods and Results should be followed! Check and correct all such discrepancies in the overall text, although they could be missed in this review.
- Ln 620, Table 3: The role of methods is to explain all features and classifiers as they are reported in Results. In fact Table 3 is NOT corresponding to the definitions in section “2.4. K Nearest Neighbor (kNN) and Deep Dense Neural Network (DDNN)”.
- Ln 637-647: This is NOT a contribution of THIS study. The overall paragraph seems UNSUITABLE for section Conclusions (at least in that informative form). Indeed, an analysis of the results in the context of already available solutions should be presented.
- Ln 653-654: The applied CANNOT be applied online due to the forward-backwards filtration, as presented in a previous comment.
Reviewer 3 Report
I think the authors adequetly addressed the reviewers' comments.
Author Response
Thank you again for the helpful comments.

Round 3
Reviewer 2 Report
The authors did their best to address my revision comments. I find them quite satisfactorily. My final comments are as follows:
- The title was changed to the bulky term “electrocardiography”. I suggest to use a more simple term “capacitive electrocardiograms” or “capacitive electrocardiogram signals”.
- Table 1, Table 2: The measurement units are missing. They are mentioned into the caption, however, they are not comprehensive. The standard consent for citing measurement units is by using []. Anyway, think about including the measurement units into the tables to be directly linked to the numbers.
- Figures 1, 2,4,5,10: The same discrepancy with measurement units as above: x-axis use [], y-axis use (). This is not publishable. Check all other tables and figures.
- Figure 6: Missing labels on y-axis.
- Figures 9, 12: Missing measurement units on y-axis.
- In overall the quality of many figures is very low and it is obvious that they are formatted with different styles and sizes of embedded text. In general, such mismatch is not suitable for qualitative publication. The embedded texts should be normally minimized unless their qualitative format is respected. The authors are responsible to provide figures, which are matching the standards for publication.
- Table 3 shows a small number of cECG records for analysis, including only 2-3 cases per group. All numbers below 30 could be considered inappropriate for statistical analysis. This should be noted as a limitation of the study in section Discussion and well communicated in the text. All such limitations should be also underlined.
- Section “2.4. K Nearest Neighbors and Deep Dense Neural Network” should be renamed to the standard notation “2.4. Classifiers”. No need to precise all details of Methods into the title. The common consent is to use generalized titles.
- I have concerns about the correctness of the word “eliminating” used in many places of the manuscript. It DOES NOT correspond to the term “suppression”, i.e. noise reduction to some level, where the noise still exists, and it is relevant to estimate signal-to-noise ratio. The term “eliminate” corresponds to the term “deletion”, i.e. exclusion part of the signal from analysis. Which one is correct: “deletion” or “suppression”? The meaning of the term should be well communicated and clarified upon its first mention in the text.
